# How to Manage Philadelphia-Positive Acute Lymphoblastic Leukemia in Resource-Constrained Settings

**DOI:** 10.3390/cancers15245783

**Published:** 2023-12-10

**Authors:** Wellington Silva, Eduardo Rego

**Affiliations:** Discipline of Hematology, Hospital das Clínicas da Faculdade de Medicina, Universidade de Sao Paulo, Sao Paulo 05403-010, Brazil; eduardo.rego@hc.fm.usp.br

**Keywords:** acute lymphoblastic leukemia, Philadelphia chromosome, tyrosine-kinase inhibitor, blinatumomab, allogeneic stem-cell transplantation

## Abstract

**Simple Summary:**

Remarkable strides have been performed in the treatment of adults diagnosed with Philadelphia-positive lymphoblastic leukemia (Ph+ ALL) through the integration of newer-generation tyrosine-kinase inhibitors and monoclonal antibodies. However, it is crucial to acknowledge that most medical centers worldwide lack access to these therapies. As a result, primary strategies employed for curing this disease continue to rely on a combination of chemotherapy and allogeneic stem-cell transplantation. Additionally, the scarcity of comprehensive literature makes it particularly challenging to provide straightforward treatment recommendations. In this narrative review, our aim is to offer a real-world perspective on the monitoring and management of Ph+ ALL patients, with an emphasis on less-resourced scenarios.

**Abstract:**

Recent studies have indicated that more than half of adult patients newly diagnosed with Ph+ ALL can now achieve a cure. However, determining the most suitable protocol for less-resourced settings can be challenging. In these situations, we must consider the potential for treatment toxicity and limited access to newer agents and alloSCT facilities. Currently, it is advisable to use less intensive induction regimens for Ph+ ALL. These regimens can achieve high rates of complete remission while causing fewer induction deaths. For consolidation therapy, chemotherapy should remain relatively intensive, with careful monitoring of the BCR-ABL1 molecular transcript and minimal residual disease. AlloSCT may be considered, especially for patients who do not achieve complete molecular remission or have high-risk genetic abnormalities, such as IKZF1-plus. If there is a loss of molecular response, it is essential to screen patients for ABL mutations and, ideally, change the TKI therapy. The T315I mutation is the most common mechanism for disease resistance, being targetable to ponatinib. Blinatumomab, a bispecific antibody, has shown significant synergy with TKIs in treating this disease. It serves as an excellent salvage therapy, aside from achieving outstanding results when incorporated into the frontline.

## 1. Introduction

The integration of tyrosine-kinase inhibitors (TKI) into frontline treatment was a breakthrough in Philadelphia-positive acute lymphoblastic leukemia (Ph+ ALL) [1]. Previously, Ph+ ALL was an almost incurable disease, with few patients able to sustain their response, even with allogeneic stem-cell transplantation (alloSCT) consolidation [2]. The combination of TKIs and chemotherapy was able to increase complete molecular remission (CMR) rates and allowed more patients to proceed to alloSCT. Furthermore, long-term remission could be achieved without alloSCT in a smaller subset [3,4,5].

Overall, Ph+ ALL is treated with TKI plus chemotherapy in most countries, followed by standard consolidation with alloSCT [1]. Several protocols have been tested over the last decades, with a wide range of intensities of cytotoxic chemotherapy combined with imatinib, dasatinib, nilotinib, and, more recently, ponatinib. Doses of TKIs have also varied across protocols, as well as the monitoring of minimal residual disease (MRD) [6]. Traditionally, in Ph+ ALL, the quantification of BCR-ABL1 transcript by real-time polymerase chain reaction (RT-PCR) has been the standard for MRD monitoring, mirroring the follow-up of chronic myeloid leukemia [6]. Nevertheless, some centers and protocols have used flow cytometry or next-generation sequencing (NGS) for MRD monitoring, depending on the availability of such methods [7,8].

More recently, the medical community has witnessed even greater achievements in Ph+ ALL, with the emergence of newer-generation TKIs, such as ponatinib, and the integration of blinatumomab, a bispecific CD3-CD19 antibody [9]. These innovative approaches have achieved unprecedented survival rates, with an estimated 2-year event-free survival (EFS) of 90% for 54 patients with newly diagnosed Ph+ ALL treated with blinatumomab and ponatinib [10,11].

Whereas most researchers at the cutting edge of Ph+ ALL are excited about this outlook, most clinicians worldwide are now challenged by these new approaches. In real life, patients from low- to middle-income countries (LMIC) have limited access to BCR-ABL1 monitoring and newer agents, such as ponatinib and blinatumomab, and might experience increased toxicities with conventional chemotherapy [12,13,14,15]. These agents are difficult to afford for public agencies in LMIC. Moreover, current trial designs and the rarity of the disease might also delay approval by regulatory agencies. Indeed, few prospective reports in Ph+ ALL have substantiated treatment in LMIC, even though adaptations are sometimes employed to improve results [13,16].

In this manuscript, we intend to provide an overview of Ph+ ALL diagnosis and management, with a special focus on approaches for resource-constrained settings, aiming to optimize outcomes in this scenario.

## 2. Diagnostic Barriers in LMIC

Patients diagnosed with B-cell phenotype lymphoblastic leukemia should undergo prompt screening for *BCR-ABL1* fusions. Ideally, this screening should be conducted using PCR or fluorescence in situ hybridization (FISH) methodologies, as they allow for a timely diagnosis of the fusion. It is essential to seek both *BCR-ABL1* transcripts, namely p190 and p210 [17]. At the diagnosis, these methodologies have similar sensitivities, being complimentary in a minority of cases where PCR can be negative or with low leukemic burden, where FISH can be suboptimal [18]. Summarily, PCR seems to be a more comprehensive method as it is more sensitive, less expensive, and allows the determination of which fusion must be tracked along the treatment [18].

Typically, treatment protocols for ALL commence with a week-long pre-phase. This pre-phase usually includes corticosteroids, either with or without cyclophosphamide. The primary goal of this pre-phase is to reduce the disease burden in the peripheral blood, as well as to address any electrolyte and coagulation abnormalities present at the time of leukemia presentation [17,19]. During this initial period, it is crucial to determine the Philadelphia chromosome status of the patients, which will guide the selection of specific treatment regimens for those who are Philadelphia chromosome-positive. Unfortunately, there are limited data available regarding the turnaround time for BCR-ABL1 testing in LMICs. Performing routine genetic testing for acute leukemia can pose challenges in certain healthcare centers [16,20,21]. A delayed detection of the Philadelphia chromosome can result in patients receiving general induction remission protocols without TKIs, and, regrettably, more intensive treatments than necessary for this patient population. Therefore, it is strongly recommended that all patients receive their Philadelphia chromosome status results within a week to avoid unnecessary, intensive induction treatments (see ‘frontline induction’). Furthermore, current literature suggests that introducing TKIs as early as possible after a diagnosis of Ph+ ALL can lead to better prognosis [22,23].

## 3. Genetic Evaluation of Ph+ ALL

Complimentary genetic evaluation through conventional karyotyping is typically available at most medical centers upon diagnosis of Ph+ ALL and can provide valuable insights. Additional cytogenetic alterations (ACAs) beyond the t(9;22)(q34;q11.2) translocation are detected in 40–60% of Ph+ ALL cases, and their prognostic significance remains a topic of debate [24]. The most commonly reported ACAs in Ph+ ALL include the gain of an extra Philadelphia chromosome or the “+der(22)t(9;22),” gain of the X chromosome, trisomy 8 or 21, high hyperdiploidy (greater than 50 chromosomes), monosomy 7, or deletions of chromosomes 7p and 9p, among others [25,26].

Whereas studies conducted in the pre-TKI era consistently demonstrated poorer survival rates in cases with ACAs, the same cannot be said for patients treated with modern regimens [26]. The prognostic impact of ACAs in Ph+ ALL largely depends on the type of alteration and the treatment protocol used. For instance, Aldoss et al. reported that among 78 adult patients with Ph+ ALL who underwent alloSCT and had available cytogenetic data, 53% had any ACA, with monosomy 7 being the most common (29%). In this cohort, primarily treated with imatinib-based regimens, a significant difference in EFS was observed (79.8% versus 39.5%, *p* = 0.01) [27]. In contrast, a study by Akahoshi et al., focused on allografted patients with Ph+ ALL, examined 1375 patients from the Japanese transplant registry. In this extensive cohort, 16.3% of individuals had ACAs, but no impact on EFS was detected. Monosomy 7 was the most common ACA and did not influence outcomes when analyzed independently. However, trisomy 8 did affect the relapse rate in this cohort (unadjusted hazard ratio (HR) = 3.02, 95% confidence interval (CI) 1.67–5.46) [28]. Another study addressing the same issue demonstrated that patients with complex Philadelphia translocations involving der(9)t(9;22) or der(22)t(9;22) had worse overall survival (OS) rates (HR 2.83, 95% CI: 1.479–5.347) [24].

Despite being cost-effective and widely accessible, conventional karyotyping is now recognized as suboptimal for examining baseline genomic alterations in Ph+ ALL. Higher-resolution techniques, such as microarray or multiplex ligation-probe analysis (MLPA), have revealed that deletions in the gene encoding Ikaros (IKZF1) on the 7p chromosome are common in Ph+ ALL (70–80%) [29,30,31]. Whereas the isolated impact of IKZF1 deletions on Ph+ ALL remains debatable, the association of this alteration with other copy-number alterations (CNA)—known as the “IKZF1-plus” subgroup—consistently demonstrates poor survival in this disease [32,33].

The IKZF1-plus subgroup is present in approximately 40% of Ph+ ALL cases and comprises patients with concurrent IKZF1 deletion and CDKN2A/B and/or PAX5 deletions [34,35]. These genomic alterations are associated with a poor prognosis, independent of the TKI used in therapy (imatinib, dasatinib, or ponatinib), and appear to be resistant to the addition of blinatumomab [35,36]. Unfortunately, these lesions require screening by multiplex platforms, which are not always available in LMICs. Limited experiences from these countries have been reported, yielding similar results [16,37]. Early identification of this subgroup may enable tailored therapy, including more effective treatments like ponatinib and blinatumomab, followed by alloSCT, which remains a recommended approach for this subset [9]. We summarized recommendations regarding upfront testing in Ph+ ALL in Table 1.

## 4. Frontline Induction in Ph+ ALL

Achieving complete remission (CR) is a prerequisite for curing Ph+ ALL. Whereas substantial efforts have been made in the past to increase the rate of MRD negativity after induction, it is now universally accepted that low-intensity induction is both safe and recommended for Ph+ ALL, especially in scenarios where early death is not negligible [38,39]. Several centers worldwide still offer conventional induction upfront, while BCR-ABL1 testing requires a wait in accordance with their practice.

Chalandon et al. reported a non-inferiority randomized trial comparing reduced-intensity chemotherapy (vincristine, dexamethasone, and imatinib) with a standard arm consisting of Hyper-CVAD plus imatinib. Due to fewer induction-related deaths, the hematologic CR rate was higher in the experimental arm (98.5% vs. 91.0%), with no differences in CMR rates [39]. Even though previous experiences with corticosteroids plus TKI have been published, this study validated this approach, even for young patients. In LMICs, this approach holds particular value as there are more reports of induction-related deaths [12,13].

During this initial phase, once again, early identification of the BCR-ABL1 fusion allows more patients to avoid intensive upfront chemotherapy, reducing early mortality [12,39]. It is worth noting that the Philadelphia chromosome is more frequent in older patients; therefore, lower-intensity protocols are often more appropriate for most patients [40]. These regimens usually combine higher doses of TKI with corticosteroids and/or vincristine. Febrile neutropenia is the most reported event, although peripheral neuropathy and liver toxicity can also occur during this phase. The use of asparaginase alongside TKI is rarely employed in adults, with few reports of increased liver toxicity associated with this combination [23].

Another variable in the induction remission equation is the initial TKI, which appears to be related to the rate of CMR after this cycle. Whereas imatinib and dasatinib seem to provide similar end-of-induction CMR rates after 1–2 cycles (15–20%) when combined with low-intensity treatment, ponatinib provides higher CMR rates when combined with intensive chemotherapy [34,41,42,43]. Whether this early outcome is related to long-term remission remains unclear, as BCR-ABL1 clearance is progressive throughout the consolidation phases and becomes more significant with prolonged exposure to TKI and chemotherapy [34,44].

The induction phase typically lasts for four weeks, and patients are managed through hospitalization. Blood transfusions and antibiotics for febrile neutropenia are routinely required. Anti-infective prophylaxis with trimethoprim-sulfamethoxazole and acyclovir is recommended, and the use of antifungal agents may also be considered [17]. It is important to note that azoles can potentially cause liver toxicity, especially when administered alongside TKIs [45].

Recent trials have integrated blinatumomab into the upfront treatment of Ph+ ALL. These regimens offer excellent CMR rates, particularly when combined with ponatinib [11,36]. Unfortunately, blinatumomab is not approved for use in frontline Ph+ ALL, and its cost makes it impractical for use in this setting in LMICs. Rituximab is also variably associated with chemotherapy in Ph+ ALL, either in patients with CD20 expression or regardless of its expression, although its benefit remains unclear [5,46,47]. In a subgroup analysis of the UKALL14 trial, which included 172 Ph+ ALL patients, the integration of four doses of rituximab in the induction phase did not demonstrate any EFS or OS benefits [46]. Please refer to Table 2 for a summary of reduced-intensity regimens for induction remission in Ph+ ALL.

## 5. Availability of TKIs

In parallel with the chronic myeloid leukemia (CML) scenario, the affordability of TKIs is another obstacle for improving results [51]. Although imatinib has lost patent protection for a long time now, making low-cost generic versions of the drug available in most LMICs, the same cannot be said about remaining TKIs. The cost per drug varies greatly among countries worldwide, especially in Latin America. Some countries may deal with higher prices for several medicines compared to those in European countries. This can be the result of ineffective or lack of pricing regulations in some countries, such Chile, Peru, and Mexico [52]. Even though imatinib has been included in the essential drug list published by World Health Organization (WHO) guidelines, it is still not provided by most governments in sub-Saharan Africa, low-income countries in Asia, or Central America [51].

When it comes to second- and third-generation TKIs, the situation is more critical in most countries, with a high rate of unaffordability pointed in pharmacoeconomical studies [52]. Imatinib generics are routinely available for USD 300–USD 3000/year in several countries, while dasatinib will be available as a generic formulation in the next few years and can be prescribed elsewhere now. The prices of patented TKIs range from USD 150,000 to USD 250,000+/year, being unaffordable even for some developed countries [53]. Moreover, the lack of clinical trials encompassing Ph+ ALL patients affects drug approvals for these patients, especially for Brazil, where only imatinib is reimbursed by the government agency.

## 6. Consolidation and Maintenance Therapy

Although CR is achieved in virtually all newly diagnosed Ph+ ALL patients, relapse rates remain high even with continued therapy, resulting in unsatisfactory long-term outcomes [6]. Previous studies in elderly populations treated with TKIs plus corticosteroids have shown that, without additional chemotherapy, leukemia progresses rapidly, indicating the need for more intensive post-remission therapy [54].

Most treatment protocols have incorporated cytotoxic chemotherapy during this phase, aiming to achieve deeper molecular remission rates and penetrate the blood–brain barrier (BBB) [5]. Retrospective data suggest that chemotherapy plays a crucial role in eradicating ABL-mutated clones, which are responsible for driving most relapses in Ph+ ALL [55,56].

The attainment of CMR is considered essential for curing Ph+ ALL [8]. As there have been few randomized trials in this disease, most current recommendations are based on the review of phase II trials and retrospective data. Outcomes obtained after these regimens indicate a complex pattern of MRD response, influenced by both the type of TKI and the associated chemotherapy.

In patients up to 18 years old, a randomized trial comparing dasatinib to imatinib showed that the former outperformed the latter with no additional toxic effects. This trial demonstrated a significant survival benefit with fewer relapses in the dasatinib arm, which can likely be extrapolated to adults [57]. A stronger benefit from ponatinib can be inferred from excellent results when combined with Hyper-CVAD, a chemotherapy backbone that has been used with different TKIs at MD Anderson over the past decades. A propensity-score matching study by Sasaki et al., comparing MRD, EFS, and OS between patients treated with ponatinib and dasatinib, consistently suggested an advantage for ponatinib [58].

Whereas integrating TKIs into chemotherapy backbones is feasible, it may lead to increased toxicity. A study by Ravandi et al. reported that 31 out of 72 patients considered eligible for the Hyper-CVAD plus dasatinib trial received fewer than the intended 8 cycles due to poor tolerance [59]. Some publications have also noted excessive toxicity during consolidation, which appears to be associated with the choice of chemotherapy prescribed in these block [12,59,60]. High doses of methotrexate (MTX) and cytarabine (HD-AraC) are typically administered, along with anthracyclines, cyclophosphamide, and clofarabine in some protocols. Rousselot et al. reported a treatment-related mortality of 12% in elderly patients treated with the EWALL-PH-01 protocol, which included alternating cycles of MTX plus asparaginase (odd courses) and HD-AraC (even courses) for six cycles during consolidation [48].

In resource-constrained settings and with the unavailability of newer drugs, adjustments in the consolidation blocks are necessary. Dose reductions, particularly in patients who achieve CMR early, should be considered. Although it is common to interrupt chemotherapy in patients with poor tolerance in clinical practice, clinicians should be aware of the increased risk of relapse in such cases. A recent report from GRAAPH-2014 demonstrated that omitting HD-AraC during consolidation was associated with excessive relapses, despite achieving non-inferior molecular response rates [50]. These findings suggest that chemotherapy reductions and discontinuations must be undertaken cautiously in Ph+ ALL, especially in fit patients and in resource-constrained scenarios. Protocols that schedule alloSCT immediately after induction, with minimal or no chemotherapy before the procedure, can be challenging in LMICs where transplantation availability is limited [12,61].

For patients not referred for alloSCT, maintenance therapy with TKIs is warranted. Initially, this may be combined with vincristine and prednisone pulses for the first 1–2 years [62]. Subsequently, patients are typically maintained with TKIs and monitored for BCR-ABL1 levels. Whereas there have been recent experiences with TKI discontinuation in selected Ph+ ALL patients with long-term remission, it is not currently recommended [63].

## 7. MRD Monitoring in Ph+ ALL

Extensive literature has provided robust evidence for the use of MRD for prognostication and guiding therapy in ALL [64]. However, in the case of Ph+ ALL, the available evidence is more limited and often contentious. In this subgroup, patients are usually monitored by RT-PCR for BCR-ABL1, in bone marrow, in parallel with other conventional methods, such as flow cytometry or the quantitative PCR of rearranged immunoglobulin (IG) genes [65]. In LMICs, most MRD studies rely on flow cytometry due to the high costs associated with NGS. The quantification of BCR-ABL1 is variably available in many centers and often requires internal validation for use [66]. Achieving a major molecular response (MMR), defined as BCR-ABL1 < 10^−3^ or <0.1%, is a initial milestone after therapy, while CMR, characterized by undetectable BCR-ABL1 in 10^4^ or <0.1%, appears to be associated with even more favorable outcomes [34,44].

Overall, it is recommended that Ph+ ALL patients undergo monitoring using both qPCR and flow or IG methods, as they complement each other. The overall concordance between BCR-ABL1 and IG qPCR is 69%, as reported by Cazzaniga et al., who observed significantly higher positivity by BCR-ABL1 in a pediatric cohort [7]. This study highlighted an important issue: while BCR-ABL1 is a valuable prognostic and monitoring tool in Ph+ ALL, its prognostic significance in patients with no detectable abnormal B cells is uncertain. In the GRAAPH-2014 trial, 38% of enrolled patients consistently showed discordant results between IG and BCR-ABL1, suggesting the persistence of non-lymphoblastic BCR-ABL1-positive cells. Further analysis of sorted cells from these discordant samples revealed the expression of BCR-ABL1 in mature B-cells, T-cells, and monocytes. Intriguingly, patients with persistent clonal hematopoiesis of BCR-ABL1 had lower relapse rates in this study [67]. Similar findings were recently reported from the MD Anderson cohort [8].

In a study from the Johns Hopkins database, encompassing 81 patients with Ph+ ALL who underwent alloSCT, subjects with MRD detected by flow cytometry but not by BCR-ABL1 on pre-transplant evaluation had poorer survival [68]. These preliminary reports suggest that changes in therapy based solely on qPCR for BCR-ABL1 should be approached with caution.

In LMICs, strategies for enhancing flow cytometry laboratories and optimizing sensitivity must be prioritized. Efforts have been made in LMICs to standardize cost-effective panels for assessing MRD in ALL [69,70,71]. We recommend that MRD assessment be performed on bone marrow samples collected after each intensive cycle or, at the very least, every 3 months [17,72]. After the end of maintenance, patients should be followed every 3–6 months for at least 5 years. Bone marrow aspirate can be considered at a frequency of up to 3–6 months for at least 5 years with MRD by flow cytometry and quantitative BCR-ABL1 [17,19]. As for B-lineage ALL, bone marrow remains the prefered source for MRD assessment, since it carries more sensitivity, regardless of the methodology [73,74,75]. Especially in patients who do not proceed with alloSCT, BCR-ABL1 testing in peripheral blood is warranted throughout their lifetime. In Ph+ ALL, recommendations regarding P190 BCR-ABL1 reinforce the role of bone marrow assessment in early stages of treatment, even though peripheral blood is a reasonable alternative in late stages or if bone marrow is not feasible [66]. An increase in BCR-ABL1 levels (typically ≥1 log) and/or the reappearance of B-cells during follow-up should trigger screening for ABL mutations and the consideration of therapy modification, if indicated [1]. It is important to note that patients with Ph+ ALL often require more chemotherapy cycles to achieve negative MRD than Ph-negative patients, typically after consolidation blocks. In most protocols, a switch to a different TKI is recommended after three cycles of chemotherapy or post-consolidation, especially in pediatric-inspired protocols [1,38].

## 8. ABL Mutations and Management of TKI

After the establishment of TKI-based therapy in Ph+ ALL, studies on relapsed disease have revealed the frequent occurrence of point mutations in the BCR-ABL1 kinase domain (KD) [1,76]. Whereas initial evidence suggests that small clones with mutations exist at the time of diagnosis in Ph+ ALL, routine upfront ABL mutation testing is not currently recommended [76,77]. This is because the presence of these mutations does not always preclude a primary response to TKIs, as chemotherapy and newer agents, such as blinatumomab, play a role in eradicating such small clones [1,17]. Short et al., using duplex sequencing, discovered a high frequency of ABL mutations at the time of diagnosis (78%), but these mutations were present at extremely low levels and did not expand at relapse in any patient [78].

However, the situation is different when it comes to assessing ABL mutations at the time of relapse or treatment failure in Ph+ ALL. In such cases, the incidence of imatinib- or dasatinib-resistant mutations increases, even when less sensitive methods like conventional Sanger sequencing are used [38]. Alternative approaches for screening ABL mutations using NGS or digital PCR are not currently recommended outside of research settings, as they do not alter treatment decisions [79]. Although there is no formal recommendation, it is widely suggested that Ph+ ALL patients undergo ABL mutation testing only when they experience a loss of molecular response or in the event of relapse. The emergence of the “gatekeeper” T315I mutation confers resistance to imatinib and dasatinib but remains susceptible to ponatinib [6].

In resource-constrained settings, ABL mutation testing may not be readily available, but it can still guide the tailored use of ponatinib, which is a more expensive TKI, in patients carrying the T315I mutation. Moreover, patients with Ph+ ALL who develop this mutation appear to have a poor prognosis regardless of undergoing alloSCT, potentially serving as a marker of clonal evolution [80]. In Table 3, there is a summary of recommendations regarding treatment and monitoring in Ph+ ALL.

## 9. CNS Prophylaxis

Recent protocols have shown that enhancing intrathecal prophylaxis can reduce the occurrence of central nervous system (CNS) relapses [81]. In most modern protocols, prophylactic irradiation (RT) is no longer employed. However, CNS-directed therapy can still be considered in cases of CNS-3 disease, particularly if cranial nerve palsy or parenchymal involvement is present [82]. In Ph+ ALL, dasatinib has demonstrated efficacy in the CNS and is recommended for patients with CNS disease [82,83].

## 10. Allogeneic Stem-Cell Transplantation and Alternative Approaches

Traditionally, alloSCT has been considered the standard consolidation therapy for Ph+ ALL in first CR [17]. However, clinical trials focused on deferring this procedure in patients who achieve CMR are lacking. Retrospective reports suggest that there is no survival benefit of alloSCT in patients who attain this level of response [39,84,85]. Ghobadi et al., in the largest cohort reported so far, demonstrated that while alloSCT was associated with a reduced relapse incidence, it did not provide advantages in terms of OS or relapse-free survival. Importantly, alloSCT was linked to a higher treatment-related mortality, even when reduced-intensity conditioning was used [84].

The decision of whether to proceed with alloSCT after achieving CR in Ph+ ALL should consider several factors: (i) clinical eligibility for the procedure, considering that many patients with this disease may be older or have comorbidities that increase toxicity; (ii) the achievement of CMR; (iii) the presence of higher-risk genetic features, such as IKZF1plus; (iv) the availability of newer-generation TKIs, like ponatinib or monoclonal antibodies (blinatumomab or inotuzumab), in the case of disease progression or loss of CMR; (v) access to the monitoring of the BCR-ABL1 transcript and MRD; And (vi) local toxicity and mortality rates associated with alloSCT, especially in LMICs [6,86,87]. Retrospective reports indicate that patients who have undergone alloSCT should receive TKI therapy post-procedure for at least 2–3 years [88,89].

Patients in CMR who are candidates to defer alloSCT should ideally undergo consolidation with intensive chemotherapy while maintaining continuous TKI exposure. These patients should be kept on TKI maintenance indefinitely [9]. Initial monitoring by bone marrow aspirate is recommended every 3 months for 1–2 years, followed by BCR-ABL1 monitoring using qPCR in peripheral blood indefinitely [17,89]. In Figure 1, a suggested management flowchart for Ph+ ALL is provided.

## 11. Conclusions

In conclusion, effectively managing Ph+ ALL in LMIC presents a multifaceted challenge that demands innovative approaches and collaborative efforts. Whereas the advanced treatments and targeted therapies available in well-resourced settings have significantly improved survival rates for Ph+ ALL patients, the disparities in access to these therapies in resource-limited regions cannot be overlooked. It is crucial for healthcare providers, policymakers, and organizations to work together to develop cost-effective strategies, improve diagnostic capabilities, and expand access to essential treatments. Overall, especially in resource-constrained scenarios, upfront low-intensity regimens are recommended, followed by intensive chemotherapy as for consolidation, always associated with TKI. Imatinib is the most cost-saving TKI, while dasatinib seems to offer deeper responses and arguably increased survival rates in children. In eligible patients, alloSCT should be carefully implemented, considering local mortality and the availability of newer lines of treatment. Optimizing existing resources and fostering international partnerships will be instrumental in bridging the gap and ensuring that Ph+ ALL patients everywhere receive the best possible care, regardless of their geographic location or socioeconomic status.

## 12. Future Directions

Recent ongoing studies incorporating blinatumomab and ponatinib upfront have yielded encouraging results in Ph+ ALL [11,36,90]. Whereas ponatinib is a potent TKI with specific activity against the T315I mutation, blinatumomab is a bispecific antibody that induces polyclonal T-cell expansion, leading to tumor apoptosis [36]. In Ph+ ALL, the expansion of CD8 T-cells along with the suppression of Treg population appears to be associated with a favorable response, making these two therapies highly synergistic [91]. The latest report from the MD Anderson group, which included 58 patients treated upfront with blinatumomab and ponatinib, showed a 90% rate of CMR and an estimated 2-year EFS of 80%, with only one patient undergoing alloSCT [90].

Whereas these therapies emerge as highly potent and chemotherapy-free strategies for treating Ph+ ALL, they are not yet available for prescription in most countries, even in the context of relapsed disease. In some countries, access to these drugs may be limited to private healthcare settings, while others may have varying levels of availability. Given the rapid pace of advancements in Ph+ ALL treatment, it is unlikely that randomized trials will be designed and mature within the next few years, as they can be time- and resource-consuming. Furthermore, once these therapies become available in developed countries, the interest in obtaining approval for use in LMICs may not be prioritized. In many countries, particularly in Latin America, public health systems may not be able to afford these new agents, underscoring the need for tailored protocols for these populations and the accelerated approval of these drugs in such settings [16,92].

## Figures and Tables

**Figure 1 cancers-15-05783-f001:**
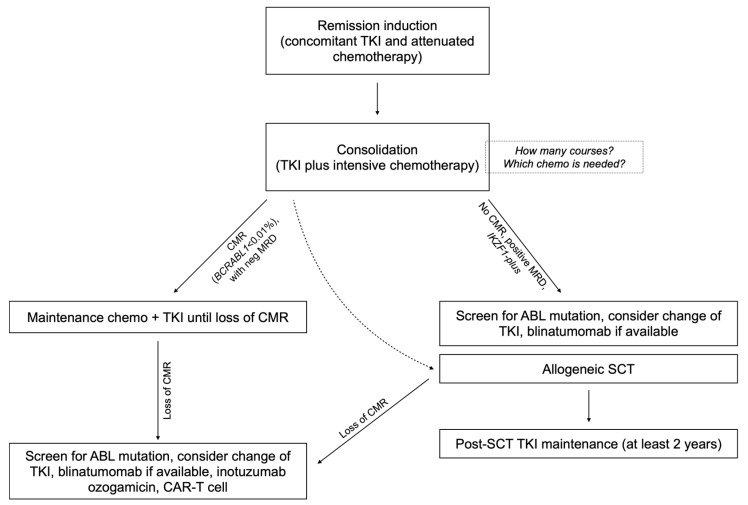
Flowchart of the suggested algorithm for the treatment of Ph+ ALL with chemo plus TKI combinations.

**Table 1 cancers-15-05783-t001:** Tests and procedures at diagnosis for a patient with suspected Ph+ ALL.

Test to Establish the Diagnosis (Minimal)	
Complete blood count and differential count Bone marrow aspirate Bone marrow trephine biopsy * Immunophenotyping by flow cytometry RT-PCR for *BCR-ABL1* (p190 and p210)	
**Genetic analyses**	**Additional tests and procedures**
Conventional karyotype MLPA for IKZF1 and correlated genes (CDKN2A/B and PAX5) Alternate methods if available: FISH for cytogenetic alterations; array for CNA alterations (e.g., IKZF1 deletions)	Complete physical examination Testicular examination, including scrotal ultrasound as indicated Performance status (ECOG/WHO score) Geriatric assessment (optional) Biochemistry, coagulation tests Hepatitis A, B, C; HIV-1 testing; CMV, EBV, HSV, VZV Serum pregnancy test Eligibility assessment for allogeneic HCT (incl. HLA-typing) Chest X-ray, 12-lead electrocardiogram, echocardiography or MUGA (on indication) Information on oocyte and sperm cryopreservation Biobanking
**Not recommended**	
Upfront ABL mutation testing	

* in patients with a dry tap.

**Table 2 cancers-15-05783-t002:** Selected trials with reduced-intensity regimens for induction remission in Ph+ ALL.

Regimen	N	Induction Regimen	Median Age (Range)	CR, %	CMR, %	Early Mortality, %
GRAAPH-2005 [39]	135	VCR 2 mg/d IV Days 1, 8, 15, and 22 DXM 20 mg/d PO Days 1–2, 8–9, 15–16, and 22–23 Imatinib 400 mg bid PO Days 1–28 Triple intrathecal Days 1, 8, and 15	48 (18–59)	98	29 (2 cycles)	0.7 (2 cycles)
GIMEMA LAL0904 [42]	51	PDN 60 mg/m^2^/d Days 1–32 Imatinib 600 mg PO Days 1–50 MTX intrathecal Days 21 and 35	46 (17–59)	96	3 (D50)	0
GIMEMA LAL1509 [34]	60	PDN 60 mg/m^2^/d Days 1–31 Dasatinib 140 mg/d PO Days 1–84 MTX intrathecal Days 0, 22, 45, 57, and 85	42 (19–59)	100	18.3	0
EWALL-PH01 [48]	71	VCR 2 mg/d IV Days 1, 8, 15, and 22 (>70 y: 1 mg) DXM 40 mg/d PO Days 1–2, 8–9, 15–16, and 22–23 (>70 y: 20 mg) Dasatinib 140 mg/d PO Triple intrathecal Days 1, 8, 15, and 22	69 (59–83)	96	20	4.2
EWALL-PH02 [49]	72	VCR 2 mg/d IV Days 1, 8, 15, and 22 (>70 y: 1 mg) DXM 40 mg/d PO Days 1–2, 8–9, 15–16, and 22–23 (>70 y: 20 mg) Nilotinib 400 mg/d bid PO Triple intrathecal Days 1, 8, 15, and 22	65 (55–85)	94	14	1.4
GRAAPH-2014 [50]	156	VCR 2 mg/d IV Days 1, 8, 15, and 22 DXM 20 mg/d PO Days 1–2, 8–9, 15–16, and 22–23 Nilotinib 400 mg bid PO Days 1–28 Triple intrathecal Days 1, 8, and 15	47 (18–60)	100	NR	2
INCB84344-201 [43]	44	PDN 60 mg/m^2^/d Days 1–29 Ponatinib 45 mg/d PO MTX intrathecal each month	66 (26–85)	91	47.7 (week 6)	4.5

VCR = vincristine; DMX = dexamethasone; PDN = prednisone; MTX = methotrexate; triple intrathecal = 15 mg MTX, 40 mg cytarabine, and 40 mg PDN; NR = not reported.

**Table 3 cancers-15-05783-t003:** Summary of the recommendations on induction, consolidation, and monitoring of Ph+ ALL.

Induction Remission (Minimal Requirements)	Induction Remission (Ideal Requirements)
Always combine TKI with chemo or low-intensity chemo Intrathecal chemotherapy Antimicrobial prophylaxis Frontline TKI: imatinib	Frontline TKI: dasatinib (improved CMR rates, potentially increased survival); ponatinib (remarkably higher survival rates). Upfront blinatumomab
**Consolidation (minimal requirements)**	**Consolidation (ideal requirements)**
Intensive chemotherapy—alternate courses, including cytarabine and methotrexate plus TKI Intrathecal chemotherapy Antimicrobial prophylaxis	Newer-generation TKI: dasatinib, ponatinib Blinatumomab for all patients or, at minimum, for those with positive MRD AlloSCT for patients with positive MRD, not achieving CMR or IKZF1-plus signature.
**Monitoring (minimal requirements)**	**Monitoring (ideal requirements)**
MRD by flow cytometry with sensitivity at minimum 10–4 (bone marrow) BCR-ABL1 by quantitative PCR ABL mutation by Sanger sequencing at relapse	MRD by IGH qPCR or NGS BCR-ABL1 by digital PCR or NGS ABL mutation by digital PCR or NGS

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
