# Peer review of "How to Manage Philadelphia-Positive Acute Lymphoblastic Leukemia in Resource-Constrained Settings"

_cancers, 2023, doi:10.3390/cancers15245783_

Round 1

Reviewer 1 Report

Dear authors, Dear editor, 
Thank you for the opportunity to read and revise this important manuscript. The topic is actual and intreaguing. However, some important points are missing. In the whole manuscript some limitations of the lower income countries should be underlined. The price and availability of the new generation TKI (in the many countries imatinib is still the only available, licensed, therapy for ALL) is one of them. Also, the difference in sensitivity, price, availability of diagnostic procedures should be underlined (e.g. FISH vs. PCR, is FISH enough). Also the importance and use of low intentsity induction is owerestimated. It is still not standard of care in the majority of centers and for that reason should be offered more as an option. 

I would like to see more practical tables and schemes comparing the standard diagnostics, minimal diagnostics (for low income countries) and ideal diagnostics.

Please define duration of MRD monitoring. 

In the whole manuscript some abbreviations are defined more then once and are not used consistently. 

Author Response

Thank you very much for taking the time to review this manuscript. Please find the detailed responses below and the corresponding revisions/corrections highlighted in the re-submitted files.

Reviewer 1

In the whole manuscript some limitations of the lower income countries should be underlined. The price and availability of the new generation TKI (in the many countries imatinib is still the only available, licensed, therapy for ALL) is one of them.

  • We added a paragraph (n. 5) underlining economic issues and availability of TKIs worldwide.

Also, the difference in sensitivity, price, availability of diagnostic procedures should be underlined (e.g. FISH vs. PCR, is FISH enough).

  • We added a sentence in section 2 addressing this issue.

Also the importance and use of low intensity induction is overestimated. It is still not standard of care in the majority of centers and for that reason should be offered more as an option. 

  • We emphasized low-intensity induction especially in scenarios where the early death is not negligible. We added a sentence highlighting this recommendation and acknowledging that most centers can still offer conventional induction regimens according to their practice.

I would like to see more practical tables and schemes comparing the standard diagnostics, minimal diagnostics (for low income countries) and ideal diagnostics.

  • Thanks for the suggestion. We added a table as requested.

Please define duration of MRD monitoring. 

  • Added.

In the whole manuscript some abbreviations are defined more than once and are not used consistently. 

  • We revised all abbreviations and corrected all of them.

Reviewer 2 Report

Well written but the title is misleading.  This is an interesting and very relevant concept for much of the world. The authors have focused on what is an ideal approach to treating this disease and then bring up what might not be available to physicians in middle income countries.  The title suggests that suggestions will be made and not a hope list which the review suggests.  Even in many high income countries, some of these things are not available.

The main thing that has to make this manuscript acceptable and useable is a suggestion list in the conclusions.  This could be broken down by topic - chemotherapy to use, TKI, monitoring, transplant etc.  In each section, there should be an absolutely needed to do things effectively and safely section, what to use if not available, a nice to have list which would make things easier to do, and finally a section on what I could do if I had all the resources.

By doing this, it would help resource-strapped programs to prioritize choices of test, drugs, etc and also when funding is requested, where to get the most for the money and most likely to be approved.  

Author Response

Thank you very much for taking the time to review this manuscript. Please find the detailed responses below and the corresponding revisions/corrections highlighted in the re-submitted files.

Well written but the title is misleading.  This is an interesting and very relevant concept for much of the world. The authors have focused on what is an ideal approach to treating this disease and then bring up what might not be available to physicians in middle income countries.  The title suggests that suggestions will be made and not a hope list which the review suggests.  Even in many high income countries, some of these things are not available.

  • We agree with this comment. However, we selected the term “resource-constrained” acknowledging that most centers have treated their Ph+ ALL far from the ideal or recommended scenario. We have made some modifications in the text and added another recommendation table on diagnostic procedures for the disease.

The main thing that has to make this manuscript acceptable and useable is a suggestion list in the conclusions.  This could be broken down by topic - chemotherapy to use, TKI, monitoring, transplant etc. 

  • We have, accordingly, added in the conclusion an assertive topic with a suggestion list as recommended by the reviewer.

In each section, there should be an absolutely needed to do things effectively and safely section, what to use if not available, a nice to have list which would make things easier to do, and finally a section on what I could do if I had all the resources.

  • Thanks for the input. We added a table summarizing diagnostic procedures as mentioned before, and another table summarizing recommendations for treatment and monitoring using a stratification, as suggested (table 3).

Round 2

Reviewer 1 Report

I think that the work has been significantly improved. 

Author Response

Thank you very much again for taking the time to review this manuscript. No comments to be addressed.        

Reviewer 2 Report

In terms of long term follow up of patients, why bone marrow.  PCR, FISH and flow can all be done on peripheral blood and cheaper than bone marrow.  In fact pcr may be more accurate on peripheral blood.  Please justify bone marrow and reference.  

Author Response

Thank you very much for taking the time to review this manuscript. Please find the detailed response below and the corresponding revisions/corrections highlighted in the re-submitted file.

In terms of long term follow up of patients, why bone marrow.  PCR, FISH and flow can all be done on peripheral blood and cheaper than bone marrow.  In fact pcr may be more accurate on peripheral blood.  Please justify bone marrow and reference.  

We added a comment on peripheral blood versus bone marrow, with suitable references in section 7 (MRD monitoring). 

Regards,

Dr. Wellington Silva and Prof. Dr. Eduardo Rego